# C-Reactive Protein to Albumin Ratio and Prognostic Nutrition Index as a Predictor of Periprosthetic Joint Infection and Early Postoperative Wound Complications in Patients Undergoing Primary Total Hip and Knee Arthroplasty

**DOI:** 10.3390/diagnostics15172230

**Published:** 2025-09-03

**Authors:** Taner Karlidag, Olgun Bingol, Omer Halit Keskin, Atahan Durgal, Baris Yagbasan, Guzelali Ozdemir

**Affiliations:** 1Department of Orthopaedic Surgery, Helios ENDO-Klinik Hamburg, 22767 Hamburg, Germany; 2Department of Orthopaedics and Traumatology, Gaziantep City Hospital, 27470 Gaziantep, Türkiye; 3Department of Orthopaedics and Traumatology, Ankara Bilkent City Hospital, 06800 Ankara, Türkiye; olgunbingol@gmail.com (O.B.); omerhalitkeskin@gmail.com (O.H.K.); durgalatahan016@gmail.com (A.D.); barisyagbasan1903@gmail.com (B.Y.); drguzelali@yahoo.com (G.O.)

**Keywords:** total joint arthroplasty, C-reactive protein to albumin ratio, prognostic nutrition index, aseptic wound complication, periprosthetic joint infection

## Abstract

**Background:** Postoperative wound complications following total joint arthroplasty (TJA) significantly impact patient outcomes and healthcare costs. Reliable preoperative biomarkers for identifying patients at increased risk are critical for optimizing patient management and reducing complication rates. This study evaluated the predictive utility of the C-reactive protein to albumin ratio (CAR) and the prognostic nutritional index (PNI) for periprosthetic joint infection (PJI) and postoperative wound complications in patients undergoing total hip arthroplasty (THA) and total knee arthroplasty (TKA). **Methods:** We retrospectively studied patients who underwent primary THA and TKA in our department from March 2019 to April 2024. The study included a total of 842 patients (568 knees and 274 hips). Preoperative blood samples were assessed for serum CRP, albumin, and total lymphocyte count, facilitating the calculation of CAR and PNI values. Patient outcomes were monitored, identifying PJI and aseptic wound complications such as persistent wound drainage, hematoma, seroma, skin erosion, and wound dehiscence within 2 weeks post-surgery. **Results:** The average follow-up time for patients was 39.2 months (range 13–73 months). PJI was significantly linked with elevated admission CAR and diminished PNI ratio (*p* < 0.001 and *p* < 0.001). ROC analysis demonstrated optimal predictive cut-off values for CAR at 3.1 (Area under curve [AUC]: 0.92, specificity 97.4%, sensitivity 92.3%) and PNI at 49.4 (AUC: 0.93, specificity 94.7%, sensitivity 91.7%). Furthermore, both CAR (Odds ratio [OR]: 3.84, 95% confidence interval [CI]: 1.6–9.1, *p* = 0.002) and PNI (OR: 21.8, 95% CI: 9–48.6, *p* < 0.001) were identified as two independent risk factors associated with the development of PJI following THA or TKA. Further subgroup analysis revealed distinct predictive thresholds for CAR and PNI according to surgical procedure type (TKA and THA), enhancing diagnostic accuracy. **Conclusions:** Preoperative admission elevated CAR and decreased PNI effectively predict PJI and postoperative wound complications in THA and TKA, supporting their utility as simple, cost-effective biomarkers in clinical practice. Incorporating CAR and PNI evaluations into preoperative assessments can enhance patient stratification and preventive strategies, thus mitigating risks and improving surgical outcomes.

## 1. Introduction

Total hip arthroplasty (THA) and total knee arthroplasty (TKA) are among the most commonly performed surgeries for end-stage osteoarthrosis and other degenerative joint diseases affecting the hip and knee. These procedures significantly enhance the quality of life and functional abilities of patients suffering from severe, persistent pain [1,2]. Furthermore, there has been an increase in the number of TKA and THA patients, a trend that is likely to persist over the next decade due to an aging population and increased life expectancy [3]. Nevertheless, despite advancements in surgical techniques and studies aimed at prevention, the incidence of postoperative wound complications and periprosthetic joint infection (PJI) remains constant among patients undergoing total joint arthroplasty (TJA) [4,5].

Assessing wound complications following TJA can be critical, as neglected or poorly managed cases may lead to serious adverse outcomes, such as PJI. Galat et al. reported that among patients requiring acute surgical intervention for wound complications, the incidence of PJI after 2 years was noted to be 6.0% [6]. Additionally, 5.3% of these patients underwent subsequent major surgical procedures, which included component resection, muscle flap coverage, or even amputation. In a related case-control study, it was found that patients requiring acute surgical intervention within 30 days to address hematomas had a 12.3% likelihood of needing further major surgical intervention within 2 years. Additionally, this cohort faced a 10.5% chance of developing a PJI [7]. Furthermore, literature suggests that patients with wound complications after TJA experience notably poorer functional and clinical outcomes [8,9]. Considering the elevated mortality rates associated with PJI, it is essential to identify individuals at risk for such complications. This approach is vital not only for mitigating adverse outcomes but also for enhancing the overall effectiveness of treatment protocols.

Recent advancements in the field of orthopaedics have led to a significant increase in the development of techniques and biomarkers aimed at predicting postoperative complications. Previous research has indicated that elevated levels of CRP and low serum albumin concentrations are significantly associated with a range of complications following TJA, including PJI [10,11]. Furthermore, there is growing interest in the CRP to albumin ratio (CAR) as a combined inflammatory biomarker that may accurately predict both PJI and delirium in the context of TJA [12,13]. Additionally, the prognostic nutritional index (PNI), which is derived from serum albumin levels and total lymphocyte count (TLC), has demonstrated a significant correlation with mortality rates among patients suffering from hip fractures, as well as higher rates of surgical site infections and poor prognosis across various cancer types [14,15]. Currently, there is an insufficient understanding of the relationship between these inflammatory biomarker combinations and the incidence of aseptic wound complications and PJI subsequent to TJA. It is hypothesized that these inflammatory biomarkers may serve as valuable instruments for predicting early postoperative wound complications and PJI following TKA and THA.

The primary objective of this study was to assess the relationship between admission CAR and PNI ratios and the incidence of postoperative aseptic wound complications and PJI following THA and TKA. The secondary objective was to identify the optimal thresholds of CAR and PNI that could effectively predict PJI in patients undergoing both THA and TKA.

## 2. Materials and Methods

### 2.1. Study Design

A retrospective review of patients who underwent THA and TKA at our tertiary referral center between May 2019 and April 2024 was performed. This study was approved by the institutional review board, and all actions were taken as a result of the most recent version of the Declaration of Helsinki. All participants gave written informed consent, which included a full description of the study. The reporting recommendations of the Strengthening the Reporting of Observational Studies in Epidemiology checklist were followed for the study of research and findings [16]. The diagnosis of PJI is conducted in strict adherence to the criteria established by the 2018 International Consensus Meetings (ICM) [17].

Patients with primary osteoarthritis of the hip or knee treated by THA or TKA were retrospectively included in the study. The exclusions were as follows: (1) diagnosis of rheumatoid arthritis or other inflammatory arthropathies, (2) history of previous joint infection, (3) patients undergoing revision arthroplasty, and (4) incomplete medical records or missing preoperative laboratory data (Figure 1). After using these criteria for inclusion, the final study group was 568 TKAs and 274 THAs with at least 1 year of clinical follow-up (Table 1). The cohort included 240 males and 602 females, with a mean age of 67.2 ± 7.8 years.

### 2.2. Patient Population

All TKAs utilized posterior-stabilized type implants, and in the majority of cases, patellar replacement was not performed. Aseptic wound complications are defined as instances in which postoperative wound healing exceeds 2 weeks without the presence of infection. These complications include persistent wound drainage (PWD), hematoma, seroma, skin erosion, and wound dehiscence. Specifically, PWD is characterized by more than 2 cm of drainage in the wound dressing beyond 72 h post-surgery [18,19]. According to our institutional protocol, aspiration was regarded as a standard procedure for patients experiencing postoperative wound complications. All aspirations were conducted under sterile conditions for all joints. The aspirates were subsequently sent for microbiological culture and cell count analysis, which included WBC count and PMN% assessment. A sample was deemed infected if it met one of two major criteria or three of five minor criteria [17].

### 2.3. Data Collection

Information of all patients, such as age, sex, type of operation (THA/TKA), side of operation, body mass index (BMI) (kg/m^2^), American Society of Anesthesiologists (ASA) score, Charlson comorbidity index (CCI), and length of stay (LOS), was collected from the electronic medical record system. Patients who experienced aseptic wound complications and PJI during the follow-up period were identified and recorded.

Preoperative laboratory data (serum levels of CRP, albumin (g/dL), and TLC (/mm^3^)) were obtained and reviewed. The PNI was calculated as [10 × serum albumin (g/dL) + 0.005 × TLC (/mm^3^)] [20]. The value of the CAR was calculated as serum CRP (mg/L)/serum albumin (g/dL) [13]. Blood sampling data were meticulously gathered from the patients’ admission blood panels, conducted immediately prior to the surgical procedure. Data were obtained according to procedure type (THA or TKA). To prevent potential bias, all of the data in this study were assessed by an independent research assistant who was not part of the operative or research team.

### 2.4. Surgical Procedure and Prevention of Infection

A tourniquet was used in all patients undergoing total knee arthroplasty. For both total knee arthroplasty and total hip arthroplasty patients, 2 g of cefazolin was administered 30 min before surgery. Preoperative skin preparation was conducted using 10% povidone iodine, followed by the application of a skin drape prior to the incision. For total knee arthroplasty, we employed a standard medial parapatellar approach, while the modified Watson-Jones approach was utilized for total hip arthroplasty. Intravenous administration of cefazolin (1 g) was performed every 8 hours on the day of surgery, following the standard protocol of our institution. Notably, none of the patients in this study had an allergy to cefazolin. A suction drain was utilized for one day post-surgery, after which it was removed. Following the removal of the drain, exercise and mobilization were promptly initiated.

### 2.5. Statistical Analysis

A post hoc power analysis was conducted using G*Power version 3.1.9.4 to determine the power of the study’s participant count. The results indicated that the study was 99% powerful with an a = 0.05. Statistical analyses were performed with SPSS (IBM Corp., version 25.0, Armonk, NY, USA). Patients were divided into two groups according to whether they had periprosthetic joint infection or not during the follow-up period, and statistical comparison was performed between groups regarding age, sex, BMI, ASA, serum CRP, serum albumin, TLC, CAR, PNI, LOS, and postoperative aseptic wound complications within 2 weeks after THA or TKA. The sensitivity, specificity, and the area under the curve (AUC) of CAR, PNI, CRP, and albumin for predicting wound complications were obtained, and the best cut-off value was determined by using the receiver operating characteristics (ROC) curve analysis. Then, to calculate the odds ratio for periprosthetic joint infection, multivariate regression analysis was performed.

Furthermore, patients were divided into two groups according to whether they had aseptic operative wound problems or not, and statistical comparison was also performed between groups, regarding age, sex, BMI, ASA, serum CRP, serum albumin, TLC, CAR, PNI, and LOS. In addition, multiple regression analysis was performed to calculate the odds ratio for the occurrence of aseptic wound problems.

Summary statistics were used to describe the data: categorical variables as frequencies and proportions, while continuous variables were presented as means with standard deviations. The normality of distribution of continuous variables was analyzed using the Kolmogorov–Smirnov test. Data that were not normally distributed were compared between groups with the Mann–Whitney *U*-test. Categorical variables were analyzed using the chi-square test. A *p*-value of less than 0.05 was considered indicative of statistical significance.

## 3. Results

The average follow-up time for patients was 39.2 months (range 13–73 months). Periprosthetic joint infections occured in 23 (4.1%) knees and 10 hips (3.6%) after a mean of 4.9 months (range, 1–22 months). Demographic features of the patients are summarized in Table 1.

### 3.1. Periprosthetic Joint Infection

No significant differences were observed between patients with and without periprosthetic joint infection in relation to demographic features, serum C-reactive protein levels, and total lymphocyte count (see Table 2). However, serum albumin was markedly lower in the patients with PJI (*p* = 0.015). Admission values of the CAR and the PNI were also significantly associated with an increased risk of developing periprosthetic joint infection (*p* < 0.001 for both). Furthermore, the incidence of aseptic wound complications within 2 weeks was considerably higher among patients who developed periprosthetic joint infection (see Table 2). Multivariate regression analysis identified positive postoperative wound problems during this period as a significant risk factor for periprosthetic joint infection (odds ratio: 6.20; 95% confidence interval [CI]: 1.74–22.13; *p* = 0.018).

The evaluation of the predictive capabilities of the admission CAR and the PNI for PJI was conducted utilizing ROC curves. The AUC for CAR in predicting PJI was 0.92, with a determined cut-off value of 3.1, yielding a sensitivity of 92.3% and a specificity of 97.4%. The cut-off for PNI was established at 49.4, with an AUC of 0.93, and corresponding sensitivity and specificity rates of 94.7% and 91.7%, respectively (Figure 2). Furthermore, both CAR ([OR]: 3.84, 95% [CI]: 1.6–9.1, *p* = 0.002) and PNI (OR: 21.8, 95% CI: 9–48.6, *p* < 0.001) were identified as two independent risk factors associated with the development of PJI following THA or TKA.

### 3.2. Procedure-Specific Analysis for PJI

Additional analysis was performed for patients who had THA and TKA separately. The AUC of CAR predicting postoperative wound complications within 2 weeks after TKA was 0.87, under the threshold of 3.01, the sensitivity and specificity were 92.6% and 90.3%, respectively, and the negative predictive value (NPV) was 95.2%, respectively. The cut-off value of PNI was 49.4, with an AUC of 0.95, and a sensitivity, specificity, and NPV of 94.7%, 90.6%, and 98.6%, respectively (Figure 3).

The optimal CAR cut-off value was 2.03 with an AUC of 0.91, a sensitivity of 93.3%, a specificity of 95.4%, and an NPV of 95.5% to predict postoperative aseptic wound complications at 2 weeks in THA patients. In addition, the AUC of PNI for the postoperative wound complications within 2 weeks in THA patients was 0.92, with the cut-off value of 48.4, sensitivity of 94.2%, specificity of 91.4%, and NPV of 97.8% (Figure 3).

### 3.3. Aseptic Wound Complications

Table 3 compares the clinical features between patients with or without postoperative wound complications. There were no significant differences in the patient demographics observed between the group with and without postoperative aseptic wound complications within 2 weeks, except for the BMI and PNI (*p* = 0.030 and *p* < 0.001, respectively). The predictive role of admission PNI for postoperative wound complications was determined by the ROC curve (Figure 4). A threshold PNI value of 49.3 was chosen, with an AUC, sensitivity, specificity, and NPV of 0.93, 91.6%, 89.5%, and 98.9%, respectively.

## 4. Discussion

In 2019, over 480,000 primary Medicare/Medicaid procedures were performed, with projections indicating a rise to 1,222,988—an increase of 254%—by 2040 [21]. This growth is driven by multiple factors, including technological, safety, and efficiency improvements, as well as a sharp increase in patient demand for TKAs and THAs, which has significantly boosted volume [21]. Despite the progress made in surgical methods and implant development, PJI continues to be one of the most prevalent and serious complications leading to failures in total joint arthroplasty. PJIs are associated with significant morbidity and mortality, creating a multifaceted burden on patients, caregivers, surgeons, healthcare facilities, and the overall health system. Conversely, early postoperative wound complications, including prolonged wound drainage, wound dehiscence, skin necrosis, and hematoma, are frequently observed in arthroplasty surgery and require careful management. These complications can lead to extended antibiotic courses and their associated complications, prolonged hospitalization, and increased economic burden [22]. A comprehensive analysis of primary and revision total knee replacement procedures based on an Arthroplasty Registry demonstrated that patients who experienced postoperative hematomas exhibited a re-operation rate for infection of 13.9%, and those with wound necrosis demonstrated a re-operation rate of 14.3% at a median follow-up of 3 years [23]. Parvizi et al. reported that patients experiencing septic failure after primary or revision TJA were 12.6 times more likely to have developed a hematoma and 16.8 times more likely to exhibit prolonged wound drainage following the initial surgery [24]. Therefore, it is of great importance to identify risk groups or to take the necessary precautions preoperatively rather than to manage complications such as PJI after they occur. Here, we conducted a study involving a comparably large cohort of patients scheduled for primary TKA and THA, providing gender-specific and procedure-specific thresholds of CAR and PNI for predicting early postoperative wound complications with high diagnostic accuracy. Given their rapidity, affordability, and wide availability, we conclude that the CAR and PNI are effective diagnostic tools for determining the likelihood of PJI and early postoperative wound complications. To the best of our knowledge, this study constitutes the first comprehensive examination of these inflammatory biomarkers within the specific contexts of procedural distribution in TJA.

Serum albumin, which serves various physiological functions, is known to diminish in response to immune system activation [25]. CRP has been extensively studied as a predictive marker in infectious settings. Moreover, it has been shown to be a prognostic indicator for mortality among hip fracture patients [26]. However, CRP will never be diagnostic on its own and can only be interpreted in conjunction with a full understanding of all other clinical and pathological results. Both CRP and albumin are synthesized in the liver and are regulated by the interleukin-6 (IL-6) pathway, with CRP experiencing a positive regulatory effect while albumin is negatively impacted by inflammation [27]. This contrasting response observed during inflammation may account for the enhanced diagnostic capability of the combination of Albumin and CRP (CAR) in previous research. This is primarily due to the synergistic effects of these markers, which facilitate the sensitive detection of abnormal immunological and inflammatory conditions in PJI. A recent investigation has revealed that preoperative levels of CAR (with a cut-off value of 1.47) are significantly correlated with an increased 30-day mortality rate following hip fracture in the elderly population [28]. Moreover, Shi et al. reported that the CAR ratio (with a cut-off value of 1.7) showed superior accuracy compared to other classical inflammatory biomarkers for diagnosing PJI in revision TJA [12]. In alignment with existing literature, this study identified a significant correlation between the CAR ratio and postoperative wound complications. Furthermore, it revealed critical thresholds applicable to both TKA and THA, which are instrumental in predicting postoperative wound complications following TJA.

The nutritional status of patients prior to surgery is a critical determinant that affects short-term outcomes in individuals undergoing arthroplasty [29,30]. Researchers have endeavored to identify serological biomarkers that can rapidly detect patients at risk of malnutrition [31]. Among these biomarkers, serum albumin is widely regarded as the most thoroughly studied and recognized indicator of nutritional status [25,32]. Numerous studies have established the potential value of albumin as both a diagnostic and predictive biomarker for various atypical immune conditions. These conditions include malignant tumors, postoperative mortality, elevated costs associated with postoperative care, and extended durations of hospital stays [32,33]. Conversely, reduced preoperative lymphocyte counts associated with aging and comorbidities significantly affect the immune response to surgical stress and trauma [34]. Furthermore, lymphocyte counts demonstrate a marked decline within hours post-surgery, and persistent postoperative lymphopenia has been independently correlated with increased mortality rates among critically ill patients undergoing emergency procedures [35]. Despite substantial biological evidence supporting the role of lymphopenia-induced inflammatory dysregulation in the onset of organ dysfunction, relevant studies specifically examining the surgical patient population remain limited.

In recent years, the PNI, which is an indirect reflection of albumin and lymphocytes, has increasingly been employed to evaluate the nutritional status of patients prior to undergoing major surgical interventions [29,36]. Hanada et al. demonstrated that a low PNI with a mean value of 48.2 ± 5.7 is correlated with negative clinical outcomes and postoperative wound complications following TJA [30]. Tunçez et al. found that a low PNI with a cut-off value of 38.4 is related to prolonged hospitalization, higher blood transfusion requirements, and an elevated risk of mortality following TJA [37]. Given that lymphocytes are crucial mediators in wound healing, tissue regeneration, and infection defense, reduced lymphocyte counts may indicate immunosuppression and impaired healing capacity [38]. Our findings indicate that a lower preoperative PNI is linked to a higher risk of PJI and postoperative wound complications after TKA and THA, showing strong sensitivity and specificity. This highlights the importance of closely monitoring patients’ nutritional health as a vital part of standard evaluations prior to surgical procedures. Furthermore, surgeons should increase their awareness of this concern and advocate for the incorporation of nutritional status indicators, such as the PNI, into standard follow-up protocols. They may also contemplate postponing procedures for patients who do not meet CAR or PNI thresholds until these indicators normalize, and consider nutritional optimization or the administration of antibiotic prophylaxis.

The strength of this study was that we conducted a procedure-specific analysis of PJI and aseptic wound complications in the current study. However, this study presents certain limitations that should be acknowledged, primarily attributable to its retrospective design. Patients with incomplete medical records and laboratory data were excluded from the analysis. This exclusion may result in the introduction of selection bias, thereby undermining the adequacy of the comparison between the patients with and without postoperative aseptic wound complications. There may be some potential variability in laboratory measurements of CRP and albumin. The laboratory evaluations conducted immediately following surgery, including assessments of serum CRP, albumin, and TLC levels, were not incorporated into the analysis. Postoperative complications may be associated with nutritional status after surgical procedures. However, this study was designed to identify preoperative factors associated with aseptic wound complications. Furthermore, our results pertain to a relatively short-term definition of wound complications (2 weeks), which may not capture delayed complications. In addition, this research was carried out at a single tertiary referral center, which might not adequately reflect the wider population, and there is a lack of external validation. Institutions with a more diverse patient demographic may demonstrate different outcomes when employing CAR and PNI during preoperative evaluations.

## 5. Conclusions

This study demonstrated that patients undergoing TKA and THA who presented with a higher CAR and a lower PNI preoperatively experienced elevated rates of PJI and early postoperative aseptic wound complications, which may result in significant adverse outcomes. Furthermore, our findings identified procedure-specific thresholds for CAR and PNI that serve as predictors of PJI and early postoperative wound complications following TJA. These results support the integration of preoperative CAR and PNI assessments as part of the preoperative evaluation, particularly for patients classified as high-risk, and furnish arthroplasty surgeons with a valuable tool for patient risk stratification. Furthermore, surgeons may contemplate delaying procedures for patients who do not meet the thresholds of CAR or PNI until these indicators reach normal levels. Alternatively, they may proceed with the surgical intervention while implementing suitable preoperative and postoperative precautions. However, to validate these promising findings, further studies employing a multicenter, prospective, and comparative design are warranted.

## Figures and Tables

**Figure 1 diagnostics-15-02230-f001:**
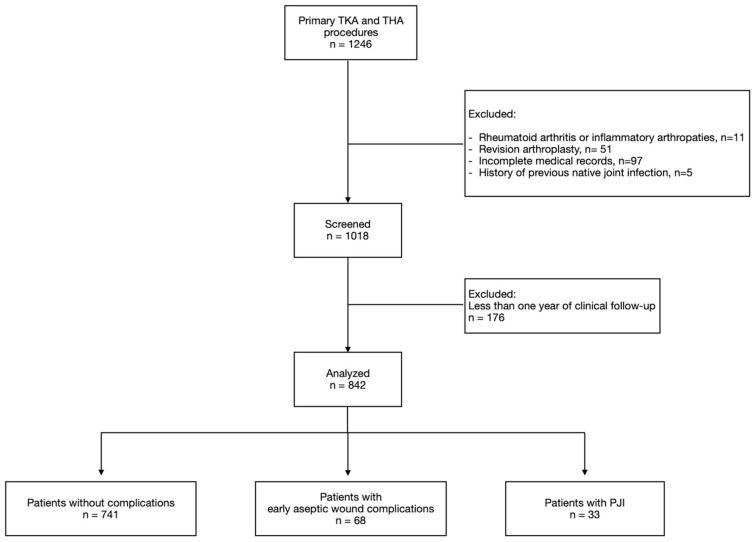
Flowchart of the study. Inclusion and exclusion criteria.

**Figure 2 diagnostics-15-02230-f002:**
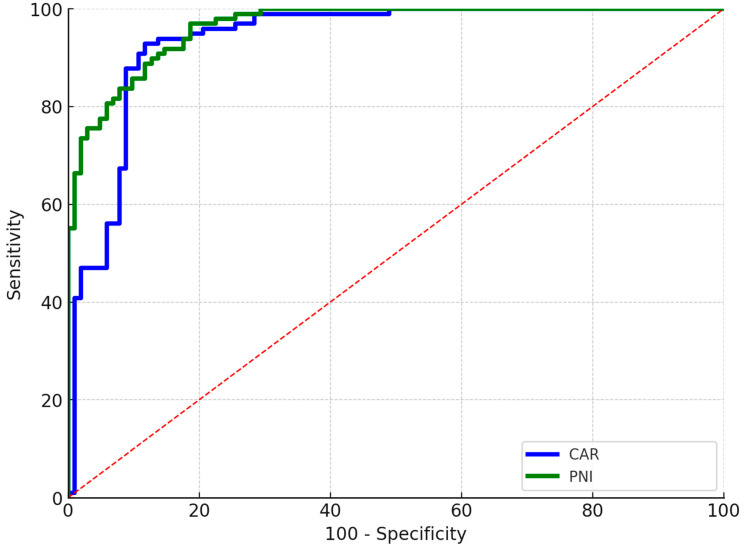
ROC curve analysis of the ability of CAR and PNI values to predict periprosthetic joint infection.

**Figure 3 diagnostics-15-02230-f003:**
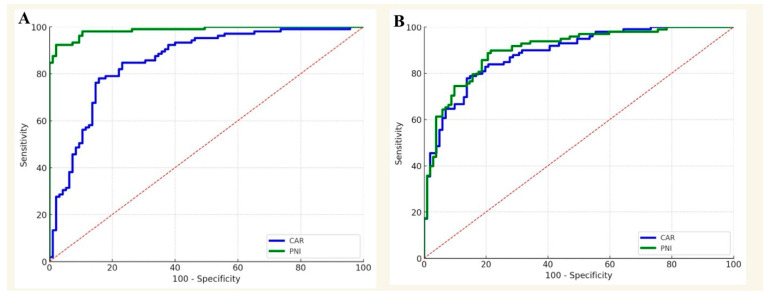
Procedure-specific ROC curve analysis of the ability of CAR and PNI values to predict periprosthetic joint infection. (**A**). Total knee arthroplasty. The AUC of CAR predicting postoperative wound complications within 2 weeks after TKA was 0.87, under the threshold of 3.01. The cut-off value of PNI was 49.4, with an AUC of 0.95. (**B**). Total Hip Arthroplasty. The optimal CAR cut-off value was 2.03 with an AUC of 0.91. The AUC of PNI for the postoperative wound complications within 2 weeks in THA patients was 0.92, with the cut-off value of 48.4. The results are presented in a 95% confidence interval.

**Figure 4 diagnostics-15-02230-f004:**
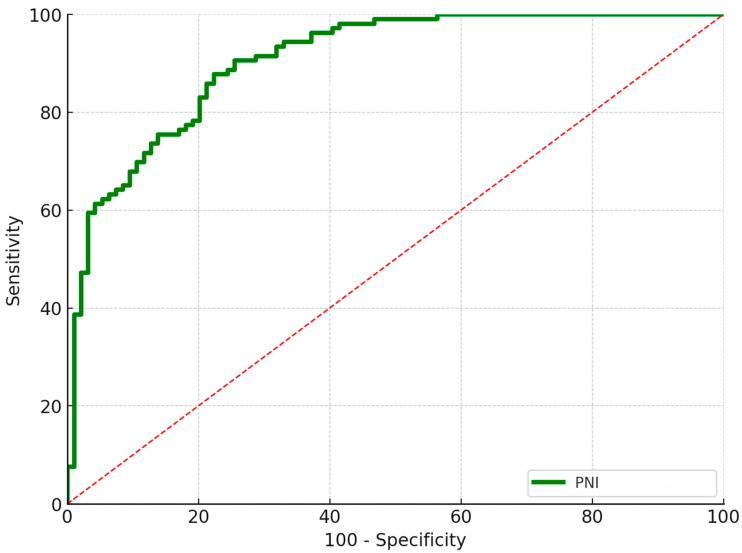
ROC curve analysis of the ability of PNI value to predict postoperative aseptic wound complications within 2 weeks. A threshold PNI value of 49.3 was chosen, with an AUC, sensitivity, specificity, and NPV of 0.93, 91.6%, 89.5%, and 98.9%, respectively. The results are presented in a 95% confidence interval.

**Table 1 diagnostics-15-02230-t001:** Demographic characteristics of the patients.

842 Patients (568 Knees, 274 Hips)	
Age (years)	67.2 ± 7.8
Sex	
Female (n, %)	602 (71.5%)
Male (n, %)	240 (28.5%)
BMI (kg/m^2^)	29.4 ± 4.5
ASA score	
ASA II (n, %)	749 (88.9%)
ASA III (n, %)	93 (11.1%)
CCI	3.59 ± 1.2
Length of stay (days)	7.58 ± 3.2
Serum CRP	3.72 ± 1.43
Serum Albumin (g/L)	4.57 ± 0.5
Total lymphocyte count (/mm^3^)	1754 ± 522
CAR	2.15 ± 0.7
PNI	54.5 ± 5.5

Values are presented as numbers, mean, standard deviation, or percentage. BMI, body mass index; ASA, American Society of Anesthesiologists score; CCI, Charlson Comorbidity Index; CRP, C-reactive protein; CAR, C-reactive protein to Albumin Ratio; PNI, prognostic nutritional index.

**Table 2 diagnostics-15-02230-t002:** Comparison between patients with PJI and patients without complications. The results are presented in a 95% confidence interval.

	Patients with PJI (n = 33)	Patients Without PJI (n = 809)	*p*
Age (years)	69.4 ± 7.8	68.5 ± 6.9	0.320
Sex (male/female)	9/24	219/590	0.842
Types of operation (THA/TKA)	13/20	232/577	0.440
BMI (kg/m^2^)	30.4 ± 5.4	29.6 ± 3.2	0.720
ASA score (ASA II/ASA III)	29/4	711/98	0.885
CCI	3.7 ± 1.8	3.6 ± 1.2	0.422
Length of stay (days)	8.9 ± 3.4	8.4 ± 2.8	0.502
Serum CRP	4.7 ± 3.3	4.1 ± 2.8	0.120
Serum Albumin (g/L)	3.9 ± 0.7	4.6 ± 0.4	0.015 *
Total lymphocyte count (/mm^3^)	1432 ± 623	1654 ± 633	0.085
CAR	3.1 ± 0.6	2.1 ± 0.8	<0.001 *
PNI	44.1 ± 4.2	55.4 ± 4.8	<0.001 *
Postoperative aseptic wound problems within 2 weeks	9 (27.2%)	59 (7.3%)	<0.001 *

Values are presented as numbers, means, standard deviations, or percentages. BMI, body mass index; ASA, American Society of Anesthesiologists score; CCI, Charlson Comorbidity Index; CRP, C-reactive protein; CAR, C-reactive protein to albumin ratio; PNI, prognostic nutritional index. *: Statistically significant.

**Table 3 diagnostics-15-02230-t003:** Comparison between patients with and without postoperative aseptic wound problems within 2 weeks. The results are presented in a 95% confidence interval.

	Patients with Aseptic Operative Wound Problems (n = 68)	Patients Without Aseptic Operative Wound Problems (n = 774)	*p*
Age (years)	69.1 ± 8.9	68.3 ± 7.6	0.240
Sex (male/female)	29/39	215/559	0.442
Types of operation (THA/TKA)	30/38	244/530	0.350
BMI (kg/m^2^)	31.1 ± 5.4	29.2 ± 4.4	0.030 *
ASA score (ASA II/ASA III)	60/8	681/93	0.805
CCI	3.9 ± 1.4	3.7 ± 1.6	0.121
Length of stay (days)	8.7 ± 5.0	8.6 ± 3.3	0.705
Serum CRP	4.3 ± 3.3	3.3 ± 2.8	0.230
Serum Albumin (g/L)	4.3 ± 0.5	4.6 ± 0.4	0.090
Total lymphocyte count (/mm^3^)	1591 ± 751	1796 ± 542	0.072
CAR	2.7 ± 0.7	2.1 ± 0.5	0.062
PNI	45.9 ± 3.9	57.3 ± 3.9	<0.001 *

Values are presented as numbers, mean, standard deviation, or percentage. BMI, body mass index; ASA, American Society of Anesthesiologists score; CCI, Charlson Comorbidity Index; CRP, C-reactive protein; CAR, C-reactive protein to albumin ratio; PNI, prognostic nutritional index. *: Statistically significant.

## Data Availability

The authors have the database available in Microsoft Excel and SPSS formats, and we are disposed to send it if it is required.

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
