# Peer review of "C-Reactive Protein to Albumin Ratio and Prognostic Nutrition Index as a Predictor of Periprosthetic Joint Infection and Early Postoperative Wound Complications in Patients Undergoing Primary Total Hip and Knee Arthroplasty"

_diagnostics, 2025, doi:10.3390/diagnostics15172230_

Round 1

Reviewer 1 Report

Comments and Suggestions for Authors

Comments

Total joint arthroplasty is an extremely successful treatment modality for the management of end-stage degenerative joint disease, particularly of the lower limb. Despite a well-performed surgery of total knee arthroplasty (TKA) and total hip arthroplasty (THA), some patients, unfortunately, have less favorable outcomes characterized by multiple different issues such as ongoing pain, swelling, instability, infection, inflammation, risk of bone fracture and arthrofibrosis. Those complications need more research effort and models to address. The authors of this manuscript have conducted retrospective investigation with large patient’s cases who underwent primary THA and TKA.  

Authors demonstrated important evidence that the patients undergoing TKA and THA who showed a higher C-Reactive Protein (CRP) and a lower Prognostic Nutritional Index (PNI) preoperatively experienced elevated rates of periprosthetic joint infection (PJI). The important contribution of this research is that the authors wave a flag that reliable preoperative biomarkers for identifying patients at increased risk are critical for optimizing patient management and reducing complication rates.

Incorporating CAR and PNI evaluations into preoperative assessments can enhance patient stratification and preventive strategies and mitigate risks and improving surgical outcomes. The manuscript is an elevated level of clinical investigation paper.

Minor suggestion:

To check the space between word and brackets throughout the manuscript. For example: Line 17: “the C-reactive protein (CRP) to albumin ratio(CAR) and the Prognostic Nutritional”.  

Line 53: Authors provided abbreviations “patients undergoing total joint arthroplasty (TJA). But in line 85, authors wrote “following total joint arthroplasty (TJA).”.

Author Response

Comments

Total joint arthroplasty is an extremely successful treatment modality for the management of end-stage degenerative joint disease, particularly of the lower limb. Despite a well-performed surgery of total knee arthroplasty (TKA) and total hip arthroplasty (THA), some patients, unfortunately, have less favorable outcomes characterized by multiple different issues such as ongoing pain, swelling, instability, infection, inflammation, risk of bone fracture and arthrofibrosis. Those complications need more research effort and models to address. The authors of this manuscript have conducted retrospective investigation with large patient’s cases who underwent primary THA and TKA.  

Authors demonstrated important evidence that the patients undergoing TKA and THA who showed a higher C-Reactive Protein (CRP) and a lower Prognostic Nutritional Index (PNI) preoperatively experienced elevated rates of periprosthetic joint infection (PJI). The important contribution of this research is that the authors wave a flag that reliable preoperative biomarkers for identifying patients at increased risk are critical for optimizing patient management and reducing complication rates.

Incorporating CAR and PNI evaluations into preoperative assessments can enhance patient stratification and preventive strategies and mitigate risks and improving surgical outcomes. The manuscript is an elevated level of clinical investigation paper.

Minor suggestion:

To check the space between word and brackets throughout the manuscript. For example: Line 17: “the C-reactive protein (CRP) to albumin ratio(CAR) and the Prognostic Nutritional”.  

Line 53: Authors provided abbreviations “patients undergoing total joint arthroplasty (TJA). But in line 85, authors wrote “following total joint arthroplasty (TJA).”.

Response: Thank you for your valuable comments and feedback. We have revised the manuscript according to your minor suggestions.

Reviewer 2 Report

Comments and Suggestions for Authors

Dear Author,

You can find my comments below,

Best regards

Title & Abstract

  • Title is clear but long; consider shortening by removing “Admission” or restructuring for conciseness.

  • Abstract is informative, but results could be reported more clearly with exact numbers and p-values to improve transparency.

  • Include the exact follow-up duration in the abstract since it is relevant to outcomes.

Introduction

  • Strengthen the rationale by emphasizing gaps in current predictive markers for PJI and wound complications.

  • Clarify novelty: highlight explicitly how your study differs from prior work on CAR, PNI, or other biomarkers.

  • Streamline background: some parts repeat concepts (CRP, albumin roles) and could be condensed.

Methods

  • Clearly specify whether this is a retrospective cohort design in the first sentence.

  • Detail sample size calculation or power analysis to justify adequacy.

  • Provide more specifics about how missing data were handled.

  • Clarify definitions: for “aseptic wound complications,” ensure the criteria are consistent with existing literature.

  • Subgroup analyses (THA vs TKA) should be pre-specified or justified to avoid post-hoc interpretation bias.

  • State blinding (if any) in laboratory assessment to strengthen methodological rigor.

Results

  • Present demographic data (Table 1) with p-values comparing groups at baseline to highlight comparability.

  • In tables, include confidence intervals for main outcomes to enhance clarity.

  • Figures (ROC curves) should include AUC, cutoffs, and confidence intervals directly on the figure or legend.

  • Procedure-specific analysis is valuable—emphasize this more strongly in text.

Discussion

  • Expand comparison with prior studies: position your cut-off values of CAR and PNI against previously published thresholds.

  • Clarify the clinical applicability: how would surgeons integrate CAR/PNI thresholds into preoperative decision-making? Would it delay surgery, trigger nutritional optimization, or guide antibiotic prophylaxis?

  • Avoid overstating: results are promising but should be framed cautiously due to retrospective and single-center design.

  • Consider potential confounders (obesity, diabetes, ASA score) more explicitly in the discussion.

Limitations

  • Already mentioned, but add:

    • Lack of external validation.

    • Possible laboratory variability in CRP/albumin measurement.

    • Relatively short-term wound complication definition (two weeks) may miss delayed complications.

Conclusion

  • Concise and aligned with results, but avoid strong recommendations about delaying surgery until validated by prospective multicenter studies.

References

  • Comprehensive, but some very recent studies (2024–2025) are included. Ensure all references cited (especially for thresholds) are up to date and relevant.

  • Verify consistency in formatting (some DOIs vs URLs). 

Author Response

Dear Author,

You can find my comments below,

Best regards

Title & Abstract

  • Title is clear but long; consider shortening by removing “Admission” or restructuring for conciseness.

Response: Thank you for pointing this out. We agree with this suggestion. Therefore we removed ‘Admission’ from the title.

  • Abstract is informative, but results could be reported more clearly with exact numbers and p-values to improve transparency.

Response: Thank you for pointing this out. We agree with this suggestion. Therefore, we have changed the results section of the abstract. You can find our revisions on page 2, lines 37-45.

  • Include the exact follow-up duration in the abstract since it is relevant to outcomes.

Response: Thank you for pointing this out. We agree with this suggestion. Therefore, we have changed the results section of the abstract. You can find our revisions on page 2, line 37.

Introduction

  • Strengthen the rationale by emphasizing gaps in current predictive markers for PJI and wound complications.

Response: Thank you for pointing this out. We agree with this suggestion. Therefore, we have changed the introduction section. You can find our revisions on page 5, lines 112-116.

  • Clarify novelty: highlight explicitly how your study differs from prior work on CAR, PNI, or other biomarkers.

Response: Thank you for pointing this out. We agree with this suggestion. Therefore, we have changed the introduction section. You can find our revisions on page 5, lines 112-116.

  • Streamline background: some parts repeat concepts (CRP, albumin roles) and could be condensed.

Response: Thank you for pointing this out. We agree with this suggestion. Therefore, we have condensed the introduction section. You can find our revisions on page 5, lines 102-116.

Methods

  • Clearly specify whether this is a retrospective cohort design in the first sentence.

Response: Thank you for pointing this out. We agree with this suggestion. Therefore, we have made some changes in the methods section. You can find our revisions on page 6, lines 130-131.

  • Detail sample size calculation or power analysis to justify adequacy.

Response: Thank you for pointing this out. We agree with this suggestion. Therefore, we have made some changes in the methods section. You can find our revisions on page 10, lines 194-196.

  • Provide more specifics about how missing data were handled.

Response: Thank you for pointing this out. We agree with this suggestion. Therefore, we have made some changes in the methods section. We already excluded the patients with incomplete medical records or missing data. You can find our flowchart on page 7. 

  • Clarify definitions: for “aseptic wound complications,” ensure the criteria are consistent with existing literature.

Response: Thank you for pointing this out. We agree with this suggestion. Therefore, we have made some changes in the methods section. We add some citations to provide consistency with the literature on page 9, lines 156-160. 

  • Subgroup analyses (THA vs TKA) should be pre-specified or justified to avoid post-hoc interpretation bias.

Response: Thank you for pointing this out. We agree with this suggestion. Therefore, we have made some changes in the methods section. We pre-specified the two distinct data collections according to procedure type on page 9, lines 176-177. 

  • State blinding (if any) in laboratory assessment to strengthen methodological rigor.

Response: Thank you for pointing this out. We agree with this suggestion. Therefore, we have made some changes in the methods section. You can find our revisions on pages 9 to 10, lines 177-179. 

Results

  • Present demographic data (Table 1) with p-values comparing groups at baseline to highlight comparability.

Response: Thank you for pointing this out. We agree with this suggestion. We already presented p-values comparing groups in tables 2 and 3 on pages 12 and 15. 

  • In tables, include confidence intervals for main outcomes to enhance clarity.

Response: Thank you for pointing this out. We agree with this suggestion. We add information about confidence intervals in tables 2 and 3. 

  • Figures (ROC curves) should include AUC, cutoffs, and confidence intervals directly on the figure or legend.

Response: Thank you for bringing this to our attention. We agree with this suggestion. We add information about the AUC, the cut-offs, and confidence intervals in the figure legends of Figures 3 and 4. 

  • Procedure-specific analysis is valuable—emphasize this more strongly in text.

Response: Thank you for pointing this out. We agree with this suggestion. Therefore we have made some changes in the manuscript. You can find our revisions on page 20, lines 386-387.

Discussion

  • Expand comparison with prior studies: position your cut-off values of CAR and PNI against previously published thresholds.

Response: Thank you for pointing this out. We agree with this suggestion. Therefore, we have made some changes in the Discussion section. You can find our revisions on page 18, lines 342-346, and on pages 19- 20, lines 370-374.  

  • Clarify the clinical applicability: how would surgeons integrate CAR/PNI thresholds into preoperative decision-making? Would it delay surgery, trigger nutritional optimization, or guide antibiotic prophylaxis?

Response: Thank you for pointing this out. We agree with this suggestion. Therefore, we have made some changes in the Discussion section. You can find our revisions on page 20, lines 378-385.

  • Avoid overstating: results are promising, but should be framed cautiously due to retrospective and single-center design.

Response: Thank you for pointing this out. We agree with this suggestion. Therefore, we avoided overstating and have made some changes in the Discussion section. You can find our revisions on page 20, lines 378-385.

  • Consider potential confounders (obesity, diabetes, ASA score) more explicitly in the discussion.

Response: Thank you for bringing this to our attention. We concur with this suggestion. However, our findings did not reveal any significant differences concerning the ASA score and diabetes within this study. Although we observed that patients with a higher BMI demonstrated a significantly increased risk of aseptic wound complications, this observation is not novel and does not add to the existing literature. Consequently, we do not consider this finding worth discussing.

Limitations

  • Already mentioned, but add:
    • Lack of external validation.

Response: Thank you for pointing this out. We agree with this suggestion. We have made some changes in the limitation part. You can find our revisions on page 21, lines 399-402.

    • Possible laboratory variability in CRP/albumin measurement.

Response: Thank you for pointing this out. We agree with this suggestion. We have made some changes in the limitation part. You can find our revisions on page 20, lines 392-393.

    • Relatively short-term wound complication definition (two weeks) may miss delayed complications.

Response: Thank you for pointing this out. We agree with this suggestion. We have made some changes in the limitation part. You can find our revisions on page 20-21, lines 397-399.

Conclusion

  • Concise and aligned with results, but avoid strong recommendations about delaying surgery until validated by prospective multicenter studies.

Response: Thank you for pointing this out. We agree with this suggestion. We have made some changes in the conclusion part. You can find our revisions on page 21, lines 411-416.

References

  • Comprehensive, but some very recent studies (2024–2025) are included. Ensure all references cited (especially for thresholds) are up to date and relevant.
  • Verify consistency in formatting (some DOIs vs URLs). 

Response: Thank you for pointing this out. We agree with this suggestion. We have reconsider the references carefully and verified consistency.